# Assessment of Genetic Diversity for Drought, Heat and Combined Drought and Heat Stress Tolerance in Early Maturing Maize Landraces

**DOI:** 10.3390/plants8110518

**Published:** 2019-11-17

**Authors:** Charles Nelimor, Baffour Badu-Apraku, Antonia Y. Tetteh, Assanvo S. P. N’guetta

**Affiliations:** 1WASCAL Graduate Research Program on Climate Change and Biodiversity, Université Felix Houphouët Boigny, Abidjan 01 BPV 34, Cote d’Ivoire; nelimor.c@edu.wascal.org; 2International Institute of Tropical Agriculture, Ibadan 200001, Nigeria; 3Department of Bioscience, Université Felix Houphouët Boigny, Abidjan 01 BPV 34, Cote d’Ivoire; nguettaewatty@gmail.com; 4Department of Biochemistry and Biotechnology, Kwame Nkrumah University of Science and Technology, University Post Office Box PMP, Kumasi 00233, Ghana; aytetteh@gmail.com

**Keywords:** climate change, combined drought and heat stress, drought, heat, landraces, maize

## Abstract

Climate change is expected to aggravate the effects of drought, heat and combined drought and heat stresses. An important step in developing ‘climate smart’ maize varieties is to identify germplasm with good levels of tolerance to the abiotic stresses. The primary objective of this study was to identify landraces with combined high yield potential and desirable secondary traits under drought, heat and combined drought and heat stresses. Thirty-three landraces from Burkina Faso (6), Ghana (6) and Togo (21), and three drought-tolerant populations/varieties from the Maize Improvement Program at the International Institute of Tropical Agriculture were evaluated under three conditions, namely managed drought stress, heat stress and combined drought and heat stress, with optimal growing conditions as control, for two years. The phenotypic and genetic correlations between grain yield of the different treatments were very weak, suggesting the presence of independent genetic control of yield to these stresses. However, grain yield under heat and combined drought and heat stresses were highly and positively correlated, indicating that heat-tolerant genotypes would most likely tolerate combined drought and stress. Yield reduction averaged 46% under managed drought stress, 55% under heat stress, and 66% under combined drought and heat stress, which reflected hypo-additive effect of drought and heat stress on grain yield of the maize accessions. Accession GH-3505 was highly tolerant to drought, while GH-4859 and TZm-1353 were tolerant to the three stresses. These landrace accessions can be invaluable sources of genes/alleles for breeding for adaptation of maize to climate change.

## 1. Introduction

Climate change is predicted to increase global temperatures and reduce rainfall patterns, with adverse effects, particularly, on the critical stages of plant growth and development, resulting in yield losses. Productivity of maize, the major staple cereal crop in sub-Saharan Africa (SSA) is thus threatened [1,2]. Rainfall under future climate change scenarios in SSA will either occur late or stop earlier than usual [1], while temperatures in large areas have already exceeded the threshold for maize growth and productivity [2,3]. Maize is highly vulnerable to drought stress (DS) and heat stress (HS) during the reproductive stages [3,4]. Drought stress at flowering and grain-filling stages of maize causes delayed silking, increased anthesis-silking interval, and reduced kernel set [5], resulting in grain yield (GY) losses above 50% [6,7]. Under extreme conditions, DS at a few days before anthesis to the beginning of grain-filling period reduced GY of maize by as much as 90% [8]. High temperatures occurring at two weeks before flowering resulted in leaf firing, increased rate of leaf senescence [9] and premature lodging [7]. At the on-set of flowering, high temperature stress caused tassel blasting, leading to reduced pollen production and viability, reduced pollination rate, shortened grain-filling period, and reduced kernel and grain weight [9,10]. Together, yield losses due to these altered morpho-physiological traits was estimated at 45% or more [11,12].

Under HS, plants open their stomata to cool their leaves by transpiration, but when plants have to keep their stomata closed to reduce water loss during combined drought and heat stress (DSHS) conditions, the leaf temperature remain high, resulting in increased yield losses compared to the separate effects of the two stresses. For example, each degree rise in temperature above 30 °C resulted in 1% reduction in maize GY under optimal growing conditions (OGC), 1.7% under DS and up to 40% or more under DSHS [1]. Meseka et al. [7] found that while DS reduced GY by 58%, DSHS reduced GY by 77%. It was demonstrated that under extreme conditions, DSHS forced farmers to abandon their farmlands [13]. To cope with these stresses, farmers will need ‘climate smart’ maize varieties with low risk of failure. To develop such maize varieties, there is the need for continuous access to a wide range of alleles that are scattered in germplasm resources, particularly, the landraces.

Landraces of maize have been selected for several years for adaptation to local conditions, harbor abundant genetic diversity for important agronomic characteristics such as, phenology, growing season, resistance to diseases and insects and tolerance to abiotic stresses, including drought and/or high temperatures [14]. The availability of such genetic diversity is critical for genetic enhancement, allowing not only increased genetic gains and improved nutritional quality, but also for broadening the genetic base of elite varieties. Maize breeding efforts over the last few decades achieved remarkable success in terms of DS tolerance [3]. However, to ensure continued gains in genetic improvement, new DS tolerant source populations are needed [3]. Moreover, in contrast to DS, research on HS and DSHS has only recently began in tropical and subtropical maize [3,7,15,16]. Identifying new sources of variation for tolerance to DS, HS and DSHS will greatly complement these breeding efforts.

The primary objective of this study was to identify landraces that combine desirable secondary traits with good levels of tolerance to DS, HS and DSHS for use in maize breeding programs as sources of variation for breeding for climate change resilience.

## 2. Results

### 2.1. Variation in Weather Conditions during the Trial Periods

In 2018, the average night and day temperatures during the period when the HS and DSHS trials were carried out ranged from 17 to 25 °C and 36 to 40 °C, respectively (Table 1). During the evaluation period in 2019, average temperatures varied from 18 to 26 °C at night and from 31 to 39 °C at daytime. There were incidences of rains after grain-filling stages in May and June, with minor effects on trials. In each year, the lowest day and night temperatures were observed at the time of planting in February, and the highest in April (Table 1). In both years, the peak temperature occurred in April (Figure 1), which coincided with the flowering and grain filling stages. During this period, night temperatures varied from 18 to 27 °C in 2018 and 23 to 29 °C in 2019, while daytime temperatures ranged from 36 to 41 °C in both years.

### 2.2. Variance and Heritability Estimates

Broad-sense heritability estimates of GY of individual trials ranged from 0.37 to 0.87 under OGC, 0.66 to 0.83 under managed drought stress (MDS), 0.72 to 0.76 under HS and 0.51 to 0.61 under DSHS (Appendix A). The combined ANOVA of the 36 early maturing maize accessions evaluated under each of the stress conditions showed highly significant (*p* < 0.001) mean squares for GY and other measured traits for year, genotypes, and genotype × year interactions. Under OGC, year, and genotypes had significant effects (*p* < 0.05 or *p* < 0.001) on all measured traits except anthesis-silking interval (ASI) and husk cover (HC) (Table 2). Results of combined analysis across two years under MDS revealed that the effects of environments and genotypes were significant (*p* < 0.05 or *p* < 0.001) for all traits except environmental effects on GY and plant aspect (PASP) (Table 2). Similarly, under HS and DSHS, the combined ANOVA showed that year and genotypes and their interaction were significant (p < 0.005 or 0.001) for GY and most other traits (Table 3). Under each evaluation condition, the genotypes contributed the largest percentage to total sum of squares when compared with other sources of variation. Consequently, repeatability values of majority of the traits were high (˃0.60).

### 2.3. Phenotypic and Genetic Correlations between Treatments, and Trait Associations

The genetic correlations between GY under HS and DSHS was very high and positive (0.94) whereas that between OGC and, HS and MDS were moderate and positive, 0.61 and 0.65, respectively. Genetic correlations observed between GY of the other treatments were very weak, ranging from −0.01 between MDS and DSHS to 0.29 for OGC and DSHS (Table 4). A similar trend was observed for the phenotypic correlations between GY of the treatments (Table 4). The phenotypic correlation between flowering traits [days to anthesis (AD), days to silking (SD) and ASI] of the different treatments were positive and ranged from very low (0.07) between ASI under MDS and HS to very high (0.84) between AD under HS and DSHS (Appendix A). The association between majority of the other traits including GY were highly significant (*p* ˂ 0.001) and positive for HS vs. DSHS.

### 2.4. Step-Wise Multiple Regression and Path Coefficient Analyses

Under MDS, the step-wise multiple regression analysis identified ear aspect (EASP) and PASP as traits with significant direct effects on GY, accounting for approximately 91% of the total variation in GY (Figure 2). of these two traits, EASP had the highest direct effect (−0.62) on yield, while PASP had the lowest direct effect on GY (−0.36). There were also two traits in the second order, including ears per plant (EPP) and ASI, each contributed indirectly to GY through EASP and PASP. Traits classified in the third order included stay green characteristic (SG), HC, SD, and plant height (PLHT). The indirect contribution of the remaining four third-order traits to yield through the second-order traits are clearly shown in Figure 2. Four traits, root lodging (RL), ear rot (EAROT), stalk lodging (SL) and ear height (EHT) were identified as the fourth-order traits with significant indirect effect on GY. While EAROT and SL contributed indirectly to GY through only PLHT and RL, EHT had indirect effects through two or more traits.

Across the HS trials, three traits (EASP, PASP and RL) were identified by the step-wise multiple regression in the first order, which explained about 86% of the total variation in GY (Figure 3). EASP had the highest direct effect on GY (−0.56), while RL was the least direct contributor to GY (−0.15). Four traits (EPP, HC, PLHT and AD) were categorized into second- order traits, each contributing indirectly to GY through one or two first order traits. The traits grouped in the third order included SD, ASI, EAROT, SG, tassel blast (TB) and EHT. Of these, ASI and EAROT had significant negative indirect effects on GY through EPP and AD. There were no fourth-order traits.

Under DSHS, only two traits, EASP and EPP were categorized as first order traits accounting for about 89% of the total variation in GY (Figure 4). While the contribution of EASP to variation in GY was negative (−0.52), that of EPP was positive (0.46). Two traits, namely SD and HC, were classified as second order traits, with each contributing indirectly to GY through the two first order traits. The traits classified into third order were AD, ASI, PASP and TB, all of which had positive indirect effects on GY. Five traits, EHT, SG, RL, SL and leaf firing (LF) were classified as fourth-order traits, each contributing to GY through two or more of the third order traits. Plant height was the only trait classified into the fifth order.

### 2.5. Abiotic Stress Strongly Affected Traits and Reduced Grain Yield Levels

Under OGC, GY averaged 3205 kg/ha (Appendix A). Grain yield was reduced to 1744 kg/ha under MDS, 1443 kg/ha under HS and to 1088 kg/ha under DSHS. Anthesis was, on average, reached at 55 days under OGC and MDS and about 60 days under HS (67) and DSHS (65). As a result of reduced water availability, ASI averaged five days under MDS. Mean ASI under HS and DSHS, was three days, similar to that recorded under OGC (2 days). Plant height was reduced by approximately 25% under MDS, 14% under HS and by 7% under DSHS. Husk cover was less affected by stress conditions as indicated by the average rating value of 4 under the different evaluation conditions. A similar trend was observed for SG and PASP. Ear aspect averaged five under OGC and MDS, and six under HS and DSHS (Appendix A). While the number of ears per plant averaged 0.81, 0.57, and 0.53 under OGC, MDS, and HS, respectively, more barren plants, reflected as reduced EPP (0.40) were observed under DSHS. As a direct consequence, average yield reduction was higher under DSHS (66.04%) compared to MDS (45.60%) and HS (54.98%).

### 2.6. Germplasm Tolerant to Abiotic Stresses

The means of GY of top 11 accessions (top 10 landraces and best check) and worst 5 landraces identified using the base index under DS, HS and DSHS are presented in Table 5. Under MDS, the index values varied from −13.66 for TZm-1441 to 14.98 for GH-3505. Top 10 landraces with positive index values yielded above 2000 kg/ha under both MDS and OGC, except TZm-1473 that yielded less (1952 kg/ha) under MDS. Nine of the top 10 performing landraces under MDS (GH-3505, TZm-1317, TZm-1307, GH-4859, GH-5756, TZm-1273, TZm-1353, TZm-1312 and TZm-1311) had a yield advantage of between 4% to 42% over the best check (Aburohemaa). Under HS, the index values ranged from −8.04 for TZm-1319 to 13.53 for Check 1 (2011 TZE-W DT STR Synthetic), whereas under DSHS, it varied from −10.85 for TZm-1277 to 13.66 for Check 1 (2011 TZE-W DT STR Synthetic). Based on index selection, the outstanding landraces under HS included GH-4859, TZm-1353, TZm-1488, TZm-1441, TZm-1466, TZm-1473, TZm-1309, TZm-1325 and TZm-1317). Of these, TZm-1353 out-yielded the best check (2011 TZE-W DT STR Synthetic) by approximately 20% while GH-4859 produced yield comparable to 2011 TZE-W DT STR Synthetic. Under DSHS, the top 10 landraces identified by the base index included GH-4859, TZm-1473, TZm-1325, TZm-1441, TZm-1466, TZm-1273, TZm-1551, TZm-1452 and TZm-1353. Two landrace accessions (GH-4859 and TZm-1353) combined high yield potential with desirable secondary traits (reduced ASI, TB and LF, and increased PHLT, SG, and EPP as well as good PASP and EASP).

### 2.7. Grouping of Accessions under Abiotic Stresses

Phylogenetic constellation plots generated from the standardized data of grain yield, plant and ear aspect scores, anthesis-silking interval, ears per plant and stay green characteristics under MDS, HS and DSHS are presented in Figure 5. Under MDS, the accessions were classified into five major groups. The number of accessions in the clusters ranged from one in cluster I to 11 in clusters II and III. The accessions of clusters I, II and III were characterized by increased ears per plant, delayed senescence, and desirable plant and ear aspects that resulted in positive base index values (Appendix A). Under HS, the accessions were grouped into two clusters, each consisting of four sub-clusters (Figure 5). The first major cluster consisted of 15 accessions, which included the three checks. Majority of the accessions in this group had high grain yields, increased ears per plant, shorter ASI and good ear aspect (Appendix A). Consequently, the base index was positive for this cluster. The second major cluster consisted of 21 landraces that were largely barren with poor ear aspect scores and low grain yield, resulting in negative base index (BI) values averaging −3.79. Similarly, under DSHS, the 36 maize accessions were clustered into three major groups (Figure 5). Cluster I consisted of 12 accessions, which included two checks whereas clusters II and III were represented by 14 and 10 accessions, respectively. Most of the accessions in cluster I were characterized by increased ears per plant, good ear aspect, high grain yield and positive base index values. In contrast, the accessions of clusters II and III were generally barren with poor ear aspect scores, low yield resulting in negative base index values (Appendix A).

## 3. Discussion

Under the prevailing and future conditions of climate change, DS, HS and DSHS stresses constitute the most important environmental factors constraining maize production in SSA [2,3]. Results of climate simulation studies showed that these stresses will most likely coincide with the flowering and grain filling stages of maize growth and development in large areas of SSA [3,12] and will result in huge yield losses on farmer’s field. The value of landraces as potential donors of beneficial alleles for breeding for climate change resilience is well-recognized [14]. Our study aimed at identifying landraces with good levels of tolerance to DS, HS and DSHS for use in maize breeding programs as potential sources of beneficial alleles for developing cultivars with enhanced resilience to climate change as well as to identify key stress-adaptive secondary traits that could lead to genetic enhancement for grain yield under DS, HS and DSHS stressed environments.

The sites selected for this study were previously used for screening maize genotypes for high levels of tolerance to DS and/or HS for climate change adaptation [4,6,7]. As shown in Table 1, the HS trials were exposed to elevated temperatures, while the DSHS trials were subjected to prolonged DS at elevated temperatures. In particular, temperatures during the reproductive stages highly exceeded the optimal threshold value of 34 °C for lowland tropical maize (Figure 1), indicating that the environments used for this study were appropriate for screening the maize germplasm for tolerance to HS and DSHS.

The significant mean squares observed for GY and most other measured traits under each evaluation condition indicated that substantial genetic variation existed among the accessions, which should facilitate selection for DS, HS and DSHS tolerance and key secondary traits conferring tolerance under the research conditions. These observations corroborated the results of Gouesnard et al. [17] who suggested the presence of significant genetic variability for tolerance to abiotic stresses in tropical maize landraces. Moreover, the presence of significant means squares of genotype by environment interaction for most of the traits indicated that the accessions varied in their responses to the stresses of the different years. These findings are consistent with the results of Meseka et al. [7].

Although broad-sense heritability estimates of GY of single trials under the stresses were moderately high, these results were consistent with previous studies of maize under abiotic stress [3,6,18]. However, as indicated by Cairns et al. [3], broad-sense heritability values of single trials can be inflated because genetic variance and genotype × trial interaction variance are confounded. The high repeatability values (60%) observed for most measured traits including GY under the stressed and non-stressed environments suggested consistency in the expression of the traits under the research conditions. These results largely provided a good indication of the performance of the accessions for breeding purposes. Similar observations were reported in maize under multiple stresses [3,19].

Grain yield observed under OGC was to some extent predictive of GY under both MDS and HS conditions as indicated by the moderately positive genetic and phenotypic correlations. These observations were in agreement with the results of previous studies on abiotic stress in maize [3,7,19,20]. The GY under OGC and DSHS had weak positive genetic and phenotypic correlations (0.29 and 0.23, respectively), suggesting the presence of independent genetic factors controlling yield potential in the two conditions. These results are consistent with the findings of earlier studies by Cairns et al. [3]. The lack of both genetic and phenotypic correlations between GY under MDS and HS as well as between MDS and HS indicated that tolerance to these stresses were modulated by different genetic mechanisms. These results are in agreement with the findings of earlier studies [3,7,18], who found that tolerance to DS was genetically distinct from tolerance to DSHS. In contrast to the results of Cairns et al. [3] however, GY observed under HS and DSHS was positive and moderately high at both the phenotypic and genetic levels. This result was in agreement with the findings of Tandzi et al. [21] who demonstrated that HS tolerant maize genotypes were most likely to be tolerant to DSHS conditions. Furthermore, the significant phenotypic correlations between the same traits under the different research conditions indicated the existence of common genetic elements governing the expression of the measured traits under the different research conditions.

The stresses applied in this study had significant negative impacts on GY and other relevant traits. This result is in agreement with the reports of Cairns et al. [3] and Trachsel et al. [18]. In particular, while the days to anthesis was on the average unaffected under MDS, silking was significantly affected, resulting in delayed ASI that affected asynchrony between male and female flowers and eventually led to reduced ears per plant. Days to anthesis was on average delayed by 10 days under DSHS and by 12 days under HS whereas, silking was delayed by 13 and 11 days under HS and DSHS, respectively. These observations could be attributed to delayed emergence owing to the severe cold due to harmattan at the time of planting. The moderate reductions in plant heights observed under HS and DSHS indicated that the plants were only affected by these stresses towards the end of the vegetative phase. Similarly, the lower average plant height observed under MDS could be attributed to the incidence of the drought stress at an early stage of plant growth and development. Indeed, the MDS was imposed during the early growth stages (25 DAP) compared to the HS and DSHS, which were imposed at 32 DAP. Traits such as EPP and EASP were most negatively affected under HS and DSHS compared to MDS. Consequently, average yield reduction under MDS was lower than that observed under HS and DSHS. Comparison of the yield loss under MDS and HS to DSHS revealed that the latter had hypo-additive effects (i.e., the effect of combined stress was higher than the individual effects but lower than their sum). These results were most probably due to the interaction of HS and DS on stomatal movements. Plants have to either close their stomata under DSHS at elevated temperature to prevent water loss or keep stomata opened to cool the leaves through transpiration [22]. Alterations in these morpho-physiological mechanisms under DSHS might have caused osmotic imbalances, resulting in the huge yield losses. These results are in agreement with the findings of previous workers who reported higher yield losses from the combined effects of DS and HS than DS and HS applied alone [7,13].

Selection based on grain yield alone under DS, HS and DSHS often limits selection gains because of its complex nature and the low heritability of the trait under stress [23]. Consequently, secondary traits that are highly heritable and are associated with GY have been widely used as selection criteria for improved yield potential under abiotic stresses. In maize for instance, significant genetic gains were reported under abiotic stresses such as low nitrogen and DS by complementing GY with key secondary traits [24]. In particular, ASI, senescence, tassel blast, ears per plant, kernel per row, tassel sterility, pollen viability, and stigma receptivity and other morpho-physiological traits, such as leaf firing were identified as key secondary traits associated with GY under DS, HS and DSHS [15,16]. In the present study, sequential multiple regression analyses revealed both ear and plant aspects, and to some extent, ears per plant and root lodging as the major determinants of yield potential, accounting for more than 85% of the observable differences in grain yield levels under the stresses. The repeatability values of these traits were moderately high. Therefore, both ear and plant aspects, ears per plant and root lodging have the potential to improve the selection efficiency for GY under the target stresses. These results partly corroborated the findings of Meseka et al. [7] who reported ear and plant aspects as well as ears per plant as the principal yield determinant traits under DS and DSHS.

An important objective of the present study was to identify landraces with outstanding performance under each of the stresses for use in maize breeding programs as sources of tolerance to the stress factors. The outstanding landraces identified by the base index under each treatment, could be invaluable sources of beneficial genes/alleles for improving DS and/or HS tolerance in elite maize germplasm. In particular, GH-3505 yielded approximately 4 tons ha^−1^ under MDS, making it interesting for use in drought-prone areas. Moreover, the accessions that yielded above the best-improved checks under each research condition should be prioritized for introgression into breeding pipelines. Of special interest for breeders will be accessions GH-4859 and TZm-1353, which combined desirable secondary traits with good levels of tolerance (positive base index values) to all the applied stresses. The fact that only two accessions were tolerant to all the target stresses was most likely due to the fact that different genetic mechanisms underlie tolerance to the three stresses applied in the present study. Furthermore, cluster analysis based on phylogenetic constellation plots largely separated tolerant accessions from their susceptible counterparts indicating that the two classes of accessions were genetically dissimilar. Such clustering patterns of maize genotypes were previously reported under DS and DSHS [7] as well as DS, HS and DSHS [21].

## 4. Materials and Methods

### 4.1. Genetic Materials

One hundred and ninety-six (196) maize landraces, representing gene pools from Burkina Faso, Ghana and Togo, were sourced from germplasm banks at International Institute of Tropical Agriculture (IITA), Nigeria and the Plant Genetics Resources Institute of Ghana. The germplasm was evaluated for phenotypic variation during the main growing season in 2017 and 2018 [25]. Thirty-three landraces, comprising six each from Burkina Faso and Ghana, and 21 from Togo, were used for this study. The landraces were selected based on the expression of the adaptive traits such as shortness and early flowering under OGC [25]. Three improved populations/varieties with enhanced tolerance to DS and/or HS, which served as checks were obtained from the Maize Improvement Program at IITA (IITA-MIP), Ibadan, Nigeria. The list of the genetic materials evaluated in this study is presented in Table 6.

### 4.2. Trial Establishment and Agronomic Management

The 36 maize accessions (33 landraces plus 3 DS and/or HS tolerant populations/varieties) were evaluated for two years under OGC, MDS, HS and DSHS. The genetic materials, experimental design, and replications were the same for all evaluation conditions. The trials were arranged in 6 by 6-alpha lattice design (incomplete design) with two replications. A plot consisted of one row, 3 m long. Rows were spaced 0.75 m apart and the distance between hills was 0.40 m. Three seeds were planted in a hill and thinned to two per stand at two weeks after planting (WAP), resulting in a final plant population density of 66,666 plants ha^−1^. Pre-emergence weeds were controlled by applying gramoxone and atrazine at the rates of 1.5 L gramoxone and 2.5 L atrazine in 200 L of water ha^-1^. Subsequently, manual weeding was done to keep trials free from weeds.

In the first experiment, the accessions were evaluated at Ikenne (6°53′ N, 3°42′ E, 60 m altitude, 1200 mm annual rainfall), Nigeria, under MDS during the dry seasons of 2017/2018 and 2018/2019. The soil type at Ikenne is Eutric nitrisol [26]. The MDS at Ikenne was achieved by using a sprinkler irrigation system that provided 17 mm of water per week up to 25 days after planting (DAP). Thereafter, the irrigation was withheld until maturity, so that the maize plants depended on stored water in the soil for growth and development.

In the second experiment, the accessions were evaluated for tolerance to HS and DSHS at Kadawa (11°39′ N, 8°27′ E, 500 m altitude), Nigeria, where drought stress at elevated temperature occur between February and June. The soil type at Kadawa is characterized as Regosols, with mainly sandy to clay loam texture [27]. At Kadawa, air temperature during the dry season often range from 33 to 45 °C [7]. This enabled establishment of trials under HS and DSHS, in which water supply was carefully controlled by a furrow irrigation system. The trials under HS and DSHS were laid in adjacent blocks, separated by 15 m to prevent spill-over of irrigation water. The trials were planted on the same day in mid-February each year, and flowering and grain-filling stages occurred in April when rainfall incidence was negligible and temperatures ranged from 36 and 41 °C, allowing for exposure of the accessions to HS and DSHS (Figure 1). A furrow irrigation system was used to supply water to the crop every four days during the first 32 DAP. Thereafter, irrigation water was withdrawn from the DSHS block while the HS block continued to receive irrigation water until physiological maturity. Irrigation was resumed one week after grain filling and was applied only once per week to avoid complete loss of DSHS trials. Meteorological data were recorded during the trial period with the aid of an automated weather station.

The same set of genetic materials were initially grown under OGC during the main growing seasons of 2017 and 2018 at Ikenne and Mokwa (9°18′ N, 5°185 4′ E, altitude 457 m, 1100 mm annual rainfall), Nigeria. The soil at Mokwa is luvisol with 0.27, 0.035 and 0.48% organic C, organic N and P content [26].

### 4.3. Traits Measured

Data were recorded on several traits at flowering including the number of days from planting to when 50% of the plants in a plot were shedding pollen and had emerged silks under DS, HS, DSHS and OGC, respectively. Anthesis-silking interval (ASI) was computed as the difference between days to 50% silking and anthesis. In addition, the number of plants showing leaf firing and tassel blasting were counted on plot basis at the flowering and grain filling stages on HS and DSHS trials and converted to percentages. Plant and ear heights (PLHT and EHT) were measured as the distance from the base of the plant to the height of the first tassel branch and the node bearing the upper ear, respectively. Root lodging (SL) (percentage of plants leaning more than 30° from the vertical), and stalk lodging (SL) (percentage broken at or below the highest ear node) were recorded as taking a few days to harvest. At 70 DAP, Plant Aspect (PASP) was scored based on the general architectural appeal of plants in a plot (standability, vigour, plant and ear height, uniformity of plants, ear placement and size, as well as disease damage and lodging) using a scale of 1 to 9, where 1 = excellent overall phenotypic appeal; 2 = very good overall phenotypic appeal; 3 = good overall phenotypic appeal; 4 = satisfactory overall phenotypic appeal; 5 = acceptable phenotypic appeal; 6 = undesirable phenotypic appeal, 7 = poor overall phenotypic appeal, 8 = very poor phenotypic appeal and 9 = completely undesirable phenotypic appeal. Similarly, husk cover (HC) was rated on a scale of 1 to 9, where 1 = husks tightly arranged and extended beyond the ear tip and 9 = exposed ears. In addition, stay green characteristic (SG) was recorded for the MDS, HS and DSHS plots at 70 DAP on a scale of 1 to 9, where 1 = 10% dead leaf area; 2 = 20% dead leaf area; 3 = 30% dead leaf area, 4 = 40% dead leaf area; 5 = 50% dead leaf area; 6 = 60% dead leaf area; 7 = 70% dead leaf area; 8 = 80% dead leaf area and 9 represented 90%–100% dead leaf area. At harvest, number of ears per plant (EPP) was obtained by dividing the total number of ears per plot by the total number of plants harvested. Ear aspect (EASP) was recorded based on general appeal of the ears without the husks (ear size and number; uniformity of size, colour and texture; extent of grain filling, insect and disease damage) using a scale of 1 to 9, where 1 = excellent (clean, uniform, large, and fully filled ears with no disease/insect damage); 2 = very good ears with no disease/insect damage and fully filled grains, one or two irregularity in cob size; 3 = good with no disease/insect damage and fully filled grains, one or two irregularity in cob size; 4= mild insect damage, no disease, fully filled grains, one or two irregularity in cob size poor; 5= mild disease/insect damage and fully filled grains, one or two irregularity in cob size, 6 = severe disease/insect damage and fully filled grains, smaller cobs, non-uniform cob size; 7 = severe disease/insect damage, scanty grain filling, few ears, non-uniformity of cobs; 8= severe disease/insect damage, scanty grain filling, very few ears and 9 = only one or no ears. In the stressed trials, harvested ears from each plot were shelled to determine the percentage grain moisture. Grain yield in kg ha^–1^ was computed from the shelled grain weight and adjusted to 15% moisture content, whereas in the experiments under OGC, harvested ears of each plot were weighed and GY was estimated based on 80% shelling percentage and adjusted to 15% moisture content.

### 4.4. Data Analysis

Data from each evaluation condition (OGC, MDS, HS and DSHS) was analysed separately. Variance components were estimated using the mixed model procedure in SAS version 9.4 [28] based on the standard linear mixed model described by Vargas et al. [29]. All effects except for genotypes were considered random and the Best Linear Unbiased Predictor (BLUP) was estimated for all measured traits following the procedure of Robinson [30].

Broad-sense heritability (H^2^) of GY of each trial was estimated using META-SAS v4 [29]. Within treatment repeatability of the traits [31] were calculated on accession-mean basis using the following formula:$$R=\frac{\sigma_{g}^{2}}{\sigma_{g}^{2}+\frac{\sigma_{ge}^{2}}{e}+\frac{\sigma_{e}}{re}}$$
where $\sigma_{g}^{2\;}$ is the genotypic variance, $\sigma_{ge}^{2}$ is the variance of genotype × environment, $\sigma_{e}$ is the residual variance; e is the number of environments, and r is the number of replicates per environment.

The phenotypic and genetic correlations between GY of the different treatments were computed following the procedure of Cooper et al. [32] as:
$$\rho_{g}=\frac{\bar{\sigma_{g\left(jj^{\prime }\right)}}}{\bar{\sigma_{g\left(j\right)}\sigma_{g\left(j^{\prime }\right)}}}$$
where $\rho_{g}$ is the genetic correlation, $\bar{\sigma_{g\left(jj^{\prime }\right)}}$ is the arithmetic mean of all pairwise genotypic covariances between environment j (say, optimum) and $j^{\prime }$ (say, drought), and $\bar{\sigma_{g\left(j\right)}\sigma_{g\left(j^{\prime }\right)}}$ is the arithmetic average of all pairwise geometric means between the genotypic variance components of the environment.

The sequential path coefficient analyses [33] was performed using the Statistical Package for Social Sciences, SPSS version 17.0 [34] to determine the secondary traits with significant contributions to GY under each research condition. A step-wise regression analysis was used to categorize the predictor traits into first, second and third order based on their individual contributions to total variation in GY with minimized multicollinearity [35]. The first step involved the regression of all the traits on GY and those with significant contributions to GY at *p* < 0.05 were identified as first order traits. Subsequently, traits that were not identified as first order traits were regressed on each of the first order traits to identify those with significant contributions to GY through each of the first order traits and were categorized as second-order traits. The procedure was repeated to identify traits in subsequent orders. The path coefficients were represented by the standardized b-values obtained from the regression analysis [33,35,36].

A base index (BI) that integrated superior grain yield, EPP, ASI, PASP, EASP, and SG was used to select the best and worst performing genotypes under each treatment [37]. Each trait was first standardized with standard deviation of 1 and a mean of zero to minimize the effect of the different scales prior to integrating into the BI. The BI was computed using the equation:$$BI=\left[\left(2\times YLD_{S}\right)+EPP-ASI-PASP-EASP-SG\right]$$
where $YLD_{S}\;$ is GY under stress, PASP is plant aspect, EASP is ear aspect, EPP is ears per plant, ASI is anthesis-silking interval and SG is the stay-green characteristic. A positive BI value indicated tolerance to the applied stress while negative values indicated susceptibility [37].

Subsequently, all traits included in the BI were used as an input file to generate a phylogenetic constellation plot, which classified the accessions into genetic groups. The phylogenetic constellation plot was performed using JMP pro 14.10 [28] based on Ward’s algorithm.

## 5. Conclusions

In conclusion, highly significant differences were observed among the accessions under each of the evaluation conditions thus, enabling identification of donors with good levels of tolerance to DS, HS and DSHS. Introgression of these outstanding donors into breeding materials will help maximize genetic gains under the stress conditions. In particular, accession GH-3505 was highly tolerant to DS while GH-4859 and TZm-1353 combined desirable secondary traits with good levels of tolerance to all the stresses. These accessions can immediately be used in tropical maize breeding programs to develop cultivars with enhanced tolerance to abiotic stress to sustain production and thus, food security in the face of climate change. The lack of correlation between the stress treatments and the poor overlap of genotypes performing well across all the applied treatments indicated that tolerance to these stresses were under independent genetic control, thus emphasizing the need to screen germplasm under each abiotic stress separately. However, the results also, showed that heat-tolerant accessions were most likely to tolerate combined drought and heat stress conditions. Identifying the genomic regions potentially underlying tolerance and the gene action controlling the stress-adaptive traits might further facilitate the introgression of these valuable landraces into breeding pipelines.

## Figures and Tables

**Figure 1 plants-08-00518-f001:**
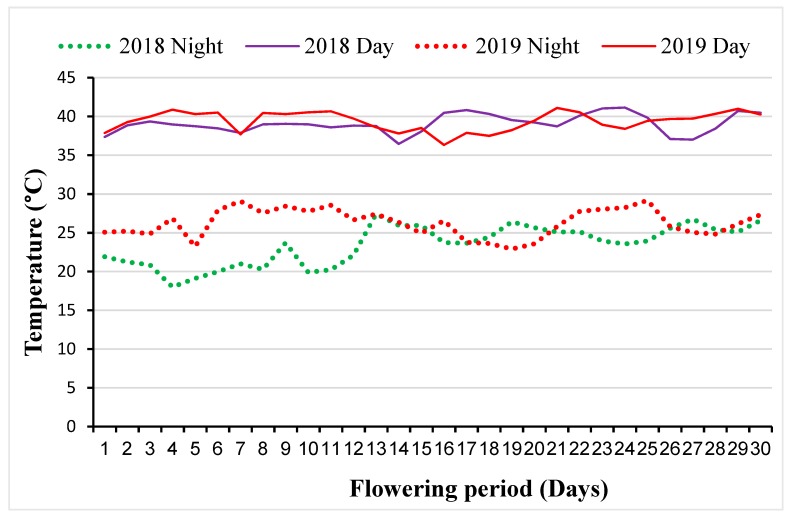
Average day and night temperatures recorded at Kadawa during the flowering period in April.

**Figure 2 plants-08-00518-f002:**
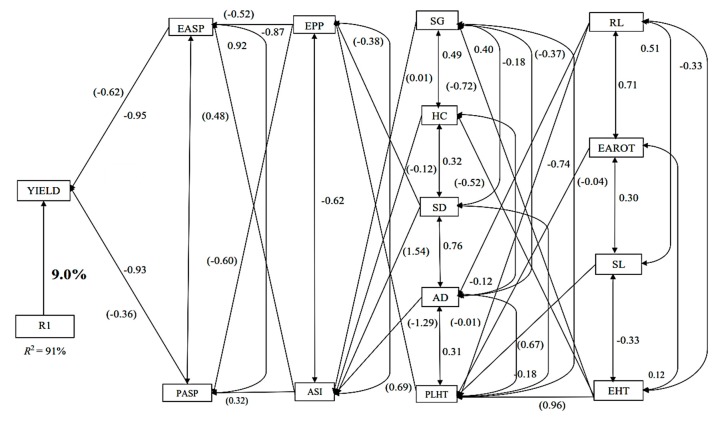
Path analysis diagram depicting the causal relationship of measured traits of the 36 maize accessions under managed drought stressed conditions. Note: Value written in bold is the error effects; the direct path coefficients are values in parenthesis and other values are correlation coefficients. R1 is error effects, R^2^ = coefficient of determination.

**Figure 3 plants-08-00518-f003:**
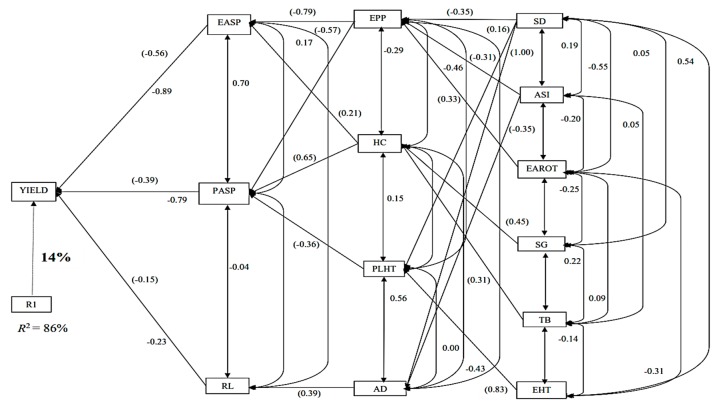
Path analysis diagram depicting the causal relationship of measured traits of the 36 maize accessions under heat stressed conditions. Note: Value written in bold is the error effects; the direct path coefficients are values in parenthesis and other values are correlation coefficients. R1 is error effects, R^2^ = coefficient of determination.

**Figure 4 plants-08-00518-f004:**
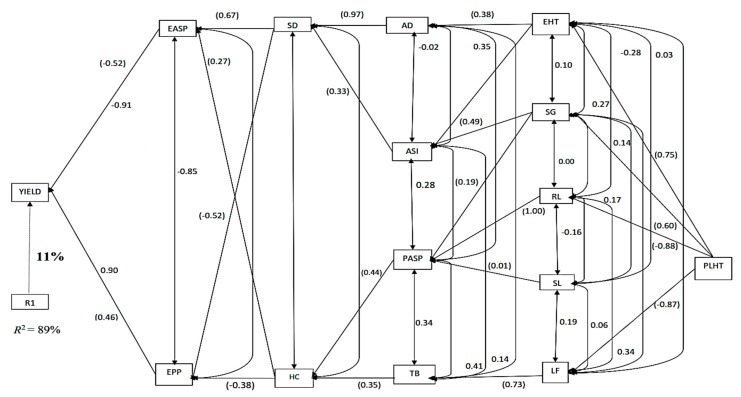
Path analysis diagram depicting the causal relationship of measured traits of the 36 maize accessions under combined drought and heat stressed conditions. Note: Value written in bold is the error effects; the direct path coefficients are values in parenthesis and other values are correlation coefficients. R1 is error effects, R^2^ = coefficient of determination.

**Figure 5 plants-08-00518-f005:**
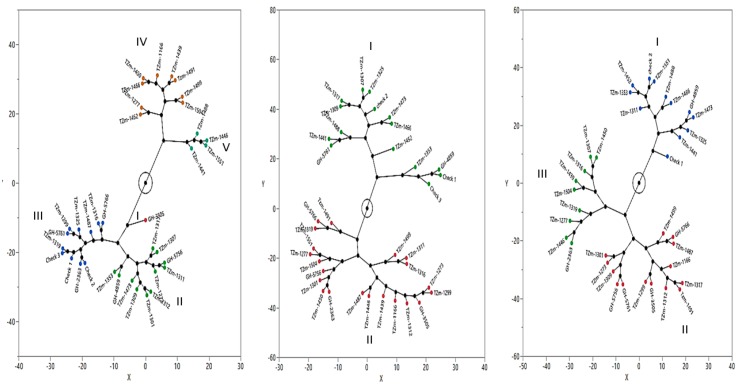
Phylogenetic constellation plots displaying the relationships between 33 maize landraces and three improved populations/varieties evaluated under managed drought stress (**left**), heat stress (**middle**) and combined drought and heat stress (**right**). Cluster I, II and III (**left**), and I (**middle** and **right**) are represented by drought, heat and combined drought and heat-tolerant accessions, respectively while the remaining clusters consisted of susceptible accessions.

**Table 1 plants-08-00518-t001:** Monthly average temperature and rainfall recorded at Kadawa, Nigeria during the trial periods in 2018 and 2019.

	2018			2019		
Month	Night (°C)	Day (°C)	Rainfall (mm)	Night (°C)	Day (°C)	Rainfall (mm)
February	17	36	0	18	31	0
March	19	38	0	24	37	0
April	24	40	0	26	39	0
May	26	39	18	26	37	37
June	25	37	47	24	34	36

**Table 2 plants-08-00518-t002:** Mean squares and repeatability values of grain yield and other traits of 36 early maturing maize accessions evaluated under optimal growing environments and managed drought stress between 2017 and 2019 in Nigeria.

Source	df	GY	AD	SD	ASI	PLHT	EHT	HC	EPP	PASP	EASP	EAROT	SG	RL	SL
**Optimal Growing Conditions**															
Env	2	2,654,514.1 *	115.6 *	198.3 **	11.7	29620.0 **	21,117.5 **	1.4	1.5 **	2.0 *	5.4 *	0.4 **	-	-	-
Rep (Env)	3	441,211.6	30.0 *	102.7 *	24.3 *	2160.6 *	1,390.5 **	0.125	0	0.4	1.7	0.1 **	-	-	-
Block (Env * Rep)	30	481,717.9 *	9	5.9	6.8	1156.6 **	405.1 *	0.79 *	0	1.1 *	1.3 *	0.0 *	-	-	-
Genotype	35	1,341,158.1 **	26.3 **	34.9 **	11.4 *	1342.1 **	627.5 **	2.2 **	0.1 **	2.0 **	3.4 **	0.0 *	-	-	-
Env *Genotype	70	216,762.9	10.3	5.3	4.5	556.5 *	311.6 *	0.5	0.0 *	0.4	0.6	0.0 *	-	-	-
Error	75	266,755.3	7.6	10.5	5.5	223.7	120.7	0.4	0	0.4	0.7	0	-	-	-
Repeatability		0.86	0.64	0.79	0.66	0.63	0.55	0.84	0.56	0.84	0.85	-	-	-	-
**Managed Drought Stress**															
Env	1	3,976,484	109.1 **	164.6 **	25.4 **	7629.7 **	9232.1 **	55.1 **	0.1 *	0.6	4.0 **	0.5 **	4.0 *	0.0 **	0.1 *
Rep (Env)	2	2,258,791	1	7.7 *	5.9 *	357.8 *	685.2 **	0.3	0	1.5 *	0.4	0.1 *	0	0	0.0 *
Block (Env * Rep)	20	1,039,093	2.1	2.6	0.7	254.4 *	100.7	0.6	0	0.5	0.3	0	0.8	0.0 *	0
Genotype	35	6,056,025.5 **	26.0 **	29.6 **	2.5 **	1957.9 **	981.2 **	96.0 **	0.1 **	2.4 **	2.7 **	0.1 **	1.4 *	0.0 **	0.0 **
Env * Genotype	35	1,840,190	4.8 **	5.6 **	1.5 *	733.2 **	342.9 **	95.8 **	0.0 *	0.7 *	0.6 *	0.0 *	0.8	0	0
Error	50	1,638,119	1.5	2.1	0.7	121.6	82.4	0.5	0	0.3	0.3	0	0.6	0	0
Repeatability		0.69	0.84	0.83	0.38	0.63	0.65	0.03	0.5	0.74	0.79	0.49	0.51	0.39	0.57

*^,^ ** Significance at 0.01 and 0.001, respectively; df: degree of freedom; Env: environment/year; Rep: replication; GY: Grain yield; AD: Days to 50% anthesis; SD: Days to 50% silking; ASI: Anthesis-silking interval; PLHT: Plant height; EHT: Ear height; HC: Husk cover; EPP: Ears per plant; PASP: Plant aspect; EASP: EAROT: Ear rot; EASP: Ear aspect; SG: Stay green characteristic; RL: Root lodging; SL: Stalk lodging; TB: Tassel blast; LF: Leaf firing.

**Table 3 plants-08-00518-t003:** Mean squares and repeatability values of grain yield and other traits of 36 early maturing maize accessions evaluated under heat stress and combined drought and heat stress during the 2018 and 2019 dry seasons at Kadawa, Nigeria.

Source	df	GY	AD	SD	ASI	PLHT	EHT	HC	EPP	PASP	EASP	SG	RL	SL	EAROT	TB	LF
		**Heat Stress**															
Env	1	57,677,836.7 **	604.3 **	855.7 **	21.8 *	28713.3 **	14600.7 **	64.0 **	2.8 **	42.3 **	134.2 **	29.3 **	-	0	0.7 **	0	-
Rep (Env)	2	4541,533.3 *	4.5	5.2	6.7	7009.3 **	2634.0 **	0.9	0.3 *	4.3 **	3	9.2 **	-	0.0 *	0.01	0	-
Block (Env * Rep)	20	768121.6	15.1 *	13.3 *	1.3	428.9 *	260.0 *	0.7 *	0.1 *	0.9 *	2.4	1.8 *	-	0	0.0 *	0	-
Genotype	35	1,836,764.3 **	27.7 **	31.2 **	4.3 *	951.3 **	495.2 **	0.7*	0.1 *	0.8 *	4.1 *	1.0 *	-	0	0.0 *	0	-
En * Genotype	35	907,223.9 *	5.6	4.9	1.8	114.6	96.8	0.5	0	0.3	1.9	0.5	-	0	0	0	-
Error	50	444,221	6	4.3	2.5	133.6	113.6	0.4	0	0.4	1.8	0.5	-	0	0	0	-
Repeatability		0.57	0.85	0.89	0.58	0.89	0.82	0.21	0.68	0.62	0.68	0.49	-	0.17	-	0.25	-
**Combined Drought and Heat Stress**																	
Env	1	6,582,075.3 **	40.1 *	242.8 **	85.6 *	14336.1 **	7764.5 **	40.1 **	0.2 *	1.6	4.7 *	0	0	0	-	0.1 **	0.1
Rep (Env)	2	1,772,416.6 **	72.4 **	148.4 **	14.3	2181.8 **	490.6 *	5.1 *	0.1	2.8 *	2.4 *	0.2	0	0	-	0.0 *	0.4
Block (Env * Rep)	20	271,106.8 *	12.2 *	12.5 *	4.3	339.0 **	254.7 **	1.4 *	0	0.6	1	1	0	0	-	0	0.1
Genotype	35	924,112.7 **	29.9 **	33.4 **	4.7	972.4 **	795.8 **	1.2*	0.1 **	1.0 *	3.3 **	1.5 *	0	0	-	0.0 **	0.1
Env * Genotype	35	405,956.3 *	7.0 *	7.9	2.9	228.3 **	171.8 **	0.7	0	0.5	1.3 *	0.6	0	0	-	0	0
Error	50	145,812.4	3.7	5.7	5.6	86	46.5	0.5	0	0.5	0.6	0.6	0	0	-	0	0
Repeatability		0.60	0.78	0.77	0.03	0.77	0.78	0.36	0.78	0.58	0.66	0.61	0.52	0.43	-	0.68	0.29

*^,^ ** Significance at 0.01 and 0.001, respectively; df: degree of freedom; Env: environment/year; Rep: replication; GY: Grain yield; AD: Days to 50% anthesis; SD: Days to 50% silking; ASI: Anthesis-silking interval; PLHT: Plant height; EHT: Ear height; HC: Husk cover; EPP: Ears per plant; PASP: Plant aspect; EASP: EAROT: Ear rot; EASP: Ear aspect; SG: Stay green characteristic; RL: Root lodging; SL: Stalk lodging; TB: Tassel blast; LF: Leaf firing.

**Table 4 plants-08-00518-t004:** Genetic correlations (down-diagonal) and phenotypic correlations (top-diagonal) between grain yield of 36 maize accessions under optimal growing conditions, managed drought stress, heat stress and combined drought and heat stress conditions.

Treatment	Optimal Growing Conditions	Managed Drought Stress	Heat Stress	Combined Drought and Heat Stress
Optimal growing conditions		0.52 **	0.45 **	0.23
Managed drought stress	0.65		0.21	−0.01
Heat stress	0.61	0.28		0.69 ***
Combined drought and heat stress	0.29	−0.01	0.94	

**^,^ *** Significant at 0.01, and 0.001, respectively.

**Table 5 plants-08-00518-t005:** Grain yield and other agronomic traits of maize accessions (best check, and top 10 and worst 5 landraces based on index selection) evaluated under managed drought stress, heat stress and combined drought and heat stress between 2017 and 2019.

Accession	GY (Kg/ha)	AD (Days)	SD (Days)	ASI (Days)	PLHT (cm)	PASP (Scale: 1–9)	SG (Scale: 1–9)	EPP	EASP (Scale: 1–9)	YR (%)	BI
**Managed Drought Stress**											
GH-3505	3901.61	56	58	3	183	3	3	0.94	3	14.75	14.98
TZm-1317	2586.92	54	57	3	180	4	3	0.79	4	34.25	8.52
TZm-1307	2336.55	54	56	2	172	5	3	0.83	5	43.38	6.57
GH-4859	2591.27	56	59	3	150	4	3	0.49	5	23.94	6.03
GH-5756	2236.72	57	60	3	163	5	3	0.80	5	50.80	5.63
TZm-1273	2251.48	56	59	3	141	5	3	0.77	5	5.70	5.50
TZm-1353	2359.25	52	56	4	153	4	4	0.64	4	44.06	5.13
TZm-1312	2462.75	57	60	4	162	5	3	0.68	5	42.47	5.10
TZm 1311	2248.39	58	62	4	173	5	3	0.75	5	23.36	4.89
TZm-1473	1951.92	49	50	2	153	5	3	0.68	5	38.71	4.67
Check 2	2139.95	52	57	5	122	5	4	0.69	5	45.29	2.33
TZm-1504	823.39	54	62	8	124	6	4	0.46	7	63.58	−6.91
TZm-1488	573.92	56	64	8	112	7	5	0.30	8	82.41	−11.89
TZm-1446	505.46	55	63	8	109	7	5	0.22	8	76.24	−12.53
TZm-1551	478.75	55	64	9	93	7	5	0.20	8	78.85	−13.21
TZm-1441	473.36	50	60	10	100	7	5	0.21	8	80.06	−13.66
**MEAN**	**1743.62**	**55**	**60**	**5**	**142**	**5**	**4**	**0.57**	**5**	**45.60**	
						**Heat Stress**					
Check 1 GH-4859	3151.11 3066.86	67 70	69 72	2 2	156 172	4 4	3 3	0.75 0.71	4 4	54.50 9.99	13.53 11.67
TZm-1353	3922.93	65	69	4	186	4	3	0.65	4	6.98	10.75
TZm-1488	1913.70	67	68	1	165	4	3	0.69	5	41.34	7.42
TZm-1441	1809.11	60	63	3	148	5	3	0.62	5	23.80	4.01
TZm-1466	2155.31	69	73	4	180	5	4	0.70	5	46.38	3.57
TZm-1473	2090.47	60	63	3	138	5	4	0.66	5	34.36	3.27
TZm-1309	1560.76	67	68	2	158	5	4	0.69	6	14.28	2.57
TZm-1325	1557.57	62	64	2	134	5	4	0.71	5	24.82	2.13
TZm-1317	1655.82	65	67	2	171	5	4	0.57	6	57.92	1.43
TZm-1273	921.25	67	71	4	160	5	4	0.44	7	61.41	−4.03
TZm-1499	494.64	72	73	1	192	5	4	0.20	8	77.52	−4.24
TZm-1299	771.25	69	73	4	170	5	4	0.34	7	62.80	−4.80
GH-5756	688.64	71	72	2	168	5	5	0.39	8	84.85	−4.87
TZm-1277	1262.86	69	72	4	168	6	5	0.36	7	62.07	−4.97
TZm-1504	750.33	64	69	4	157	6	4	0.35	7	66.81	−7.16
TZm-1319	666.63	66	71	5	180	6	4	0.35	8	79.96	−8.04
**MEAN**	**1443.00**	**67**	**70**	**3**	**162**	**6**	**4**	**0.53**	**6**	**54.98**	
**Combined Drought and Heat Stress**											
Check 1 GH-4859	2581.69 1965.05	64 69	65 71	1 2	175 152	4 5	3 4	0.95 0.74	4 5	62.72 42.32	13.66 7.37
TZm-1473	1950.12	60	63	3	150	5	4	0.64	5	38.77	6.13
TZm-1325	1783.11	60	62	2	145	5	4	0.50	5	13.93	5.20
TZm-1441	2164.05	63	66	4	157	6	4	0.66	5	8.84	5.19
TZm-1466	1807.08	67	70	3	179	4	4	0.72	5	55.04	5.09
TZm-1273	1162.94	67	68	1	201	5	3	0.48	5	51.29	3.83
TZm-1551	1554.25	62	66	3	165	5	3	0.56	5	31.34	3.58
TZm-1452	1534.90	57	61	4	168	5	3	0.50	5	28.37	3.50
TZm-1353	1796.76	62	66	4	189	4	3	0.46	5	57.40	3.27
TZm-1488	1285.44	65	68	3	173	4	3	0.44	6	60.60	1.82
TZm-1316	740.47	65	68	3	188	5	6	0.23	7	80.16	−4.55
GH-3505	565.45	68	72	4	166	5	3	0.07	8	87.64	−5.46
TZm-1439	566.12	69	74	5	178	7	4	0.21	8	67.20	−6.40
TZm-1504	512.65	66	70	4	167	6	5	0.16	8	77.33	−6.61
TZm-1277	0.00	67	71	5	172	6	4	0.00	8	100	−10.85
**MEAN**	**1088.39**	**65**	**68**	**3**	**175**	**6**	**4**	**0.40**	**6**	**66.04**	

GY: Grain yield; AD: Days to 50% anthesis; SD: Days to 50% silking; ASI: Anthesis-silking interval; PLHT: Plant height, PASP: EPP: Ears per plant; Plant aspect; PLHT: SG: Stay green characteristics; EASP: Ear aspect; YR: Yield reduction; BI: base index. Check 1: 2011 TZE-W DT STR Synthetic; Check 2: Aburohemaa.

**Table 6 plants-08-00518-t006:** List of the 36 maize accessions that were evaluated for tolerance to drought, heat and combined drought and heat stress between 2017 and 2019 at Ikenne and Kadawa, Nigeria.

No.	Accession	Origin	No.	Accession	Origin	No.	Accession	Origin
1	TZm-1273	Togo	13	TZm-1353	Togo	25	TZm-1491	Burkina Faso
2	TZm-1277	Togo	14	TZm-1439	Togo	26	TZm-1499	Burkina Faso
3	TZm-1299	Togo	15	TZm-1441	Togo	27	TZm-1504	Burkina Faso
4	TZm-1301	Togo	16	TZm-1446	Togo	28	GH-2363	Ghana
5	TZm-1307	Togo	17	TZm-1450	Togo	29	GH-3505	Ghana
6	TZm-1309	Togo	18	TZm-1452	Togo	30	GH-4859	Ghana
7	TZm-1311	Togo	19	TZm-1466	Togo	31	GH-5756	Ghana
8	TZm-1312	Togo	20	TZm-1473	Togo	32	GH-5761	Ghana
9	TZm-1316	Togo	21	TZm-1551	Togo	33	GH-5766	Ghana
10	TZm-1317	Togo	22	TZm-1166	Burkina Faso	34	2011 TZE-W DT STR Synthetic (Check 1)	IITA-MIP
11	TZm-1319	Togo	23	TZm-1487	Burkina Faso	35	Aburohemaa (Check 2)	IITA-MIP
12	TZm-1325	Togo	24	TZm-1488	Burkina Faso	36	2015 TZE-W DT STR Syn C0 (Check 3)	IITA-MIP

IITA-MIP: Maize Improvement Program at the International Institute of Tropical Agriculture.

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
