# Peer review of "Assessment of Genetic Diversity for Drought, Heat and Combined Drought and Heat Stress Tolerance in Early Maturing Maize Landraces"

_plants, 2019, doi:10.3390/plants8110518_

Round 1
Reviewer 1 Report
The manuscript is well written and gives novel information to the readers, hence I recommend this manuscript to be published from the journal "Plant". I recommend to the authors to explain the technical terms "Plant Aspect (PASP)" and "Ear Aspect (EASP)". Since I am not maize breeder, I do not know these technical terms and I can not understand even after I read the explanation written in Materials and Method. PASP and EASP are revealed to be very important secondary traits to estimate the drought and heat tolerance of materials (based on the results of this study) and therefore need more detailed explanation.
Author Response
The traits, PASP and EASP have been described in detail in the revised MS.

Reviewer 2 Report
In this manuscript author asses the genetic diversity and identify drought, heat, and its combination tolerant maize Landraces. They have used 33 varieties for this study, which is scientifically Ok. However, my concern is the lack of enough results to back this study, and just agronomic data will not be enough. There I would suggest authors to add physiological data for stress tolerance backed by morphological images. Also paper is loosly written, and much grammatical error with some random words appears all the time. Consider checking from English Expert.
Consider changing the title. It is plagiarized from another study.
Author Response
Standard management of drought and heat stress trials in field phenotyping (Badu-Apraku et al., 2012; Zaidi et al., 2016; 2019) were followed in this study. Moreover, the present study utilized more traits compared to previous studies that aimed at identifying drought and heat tolerant maize germplasm (Cairns et al., 2013; Meseka et al., 2018). In addition, data on stay green, leaf firing and tassel blast, which were included in the present study are important physiological indicators of stress tolerance in maize. Therefore, we are fully convinced that the data utilized was enough to back the results of the study. Grammatical errors, to the best of our knowledge have been corrected in the revised MS. The title have been modified to read ‘Assessment of Genetic Diversity for Drought, Heat and Combined Drought and Heat Stress Tolerance in Early Maturing Maize Landraces’.Reviewer 3 Report
The manuscript is on the diversity study for the drought and high temperature stress tolerance and their combination in maize.
The topic is important to develop the maize varieties tolerant to dry and high temperature stress condition in the flowering times. However, I felt difficult to follow the contents because the storyline starts from the heritability without showing the actual distribution of the original populations (actually, the general statistics was explained a little near at the end of the manuscript). I suggest that the authors should start to explain the distribution of the population and the general statistics. I couldn't find the statistics of the whole populations.
Heritability could be explained and compared among different conditions easier way of presentation. I suggest that the proper GxE analysis using AMMI would be better to explain your data. In the case, table 5 is not needed. It seems not that attractive to see the data of the lines - I could't assess if the data of the table is proper.
Four lines in Fig.1 should be shown differently. They look alike. Table 3 should be aligned better.
For figure 5, phylogenetic diagram is proper for the genetic similarity and implies an evolutionary relationship. I don't think that any information for the genetic-evolutionary relationship of the varieties were provided. I'm not sure if individual phenotypic values of each trait can be the source of the genetic value. They are not independent.
I suggest that the authors should revise the manuscript properly and submit again.
Author Response
The general distribution of the genotypes including the general statistics are based on data from individual trials that were performed during the study period. Previous studies (Badu-Apraku et al., 2017) underpinned the need to eliminate data of trials with low broad sense heritability for grain yield (less than 0.30) since such trials are thought to be highly influenced by environment and hence coukd be misleading. Therefore, we started the storyline with the heritability of grain yield of individual trials to give credence to subsequent results that were discussed. The idea was not to compare the heritability of the different managements as suggested. The objective of this study was to identify landrace accessions with high grain yield potential and desirable secondary traits for use for improving drought and/ or heat stress tolerance in maize. It was necessary to use an index that integrates grain yield with priority traits (important secondary traits) as selection criterion. The focus was not to identify stable genotypes. Therefore, GxE analysis using AMMI to the best of our knowledge will not appropriately describe the data. To the best of our knowledge, Table 5 is necessary because it highlights the outstanding and worse genotypes identified using an index selection that integrated high grain yield with five other priority traits, which was the main focus of the study. Figure 1 has been formatted to make it readable. Table 3 has been properly aligned in the revised MS. Please, the genotypes evaluated in this study were landraces from different countries and not varieties. The phylogenetic trees did not cluster the accessions according to their origin. The genetic relationships observed were largely due to their response to the applied stresses in which tolerant genotypes were separated from their susceptible counterparts. Therefore, it was not feasible to provide information on the genetic-evolutionary relationship as suggested.Reviewer 4 Report
The manuscript reports on an assessment varieties and traits conducive with drought and heat tolerance in maize.
The introduction provides an interesting overview to the topic identifying the key components of the crop life-cycle that are vulnerable to drought and heat stress and potential reasons with reference to plant physiology.
Specific comments: avoid abbreviations in the abstract, also definitions to abbreviations need to be provided in the introduction irrespective of whether previously defined in the abstract.
page 3 line 58 what is DSHTS an abbreviation of?
The length of the results section would benefit from the text being shortened and not listing off table contents in places e.g. page 6 line 113 onwards 'Grain yield under OGC was moderately and positively correlated with GY under MDS (0.65) and HS (0.61) (Table 4). Grain yield under OGC was weakly and positively correlated with GY under DSHS (0.29). Grain yield observed under HS was not predictive of GY under MDS (0.28). There was no significant relationship between GY under MDS and DSHS (-0.01). The genetic correlation observed between GY under HS and DSHS was positive and very high (0.94). The phenotypic correlation...'. Draw out the significant points.
The text on page 29 of the discussion line 348-359 lists the top 10 landraces. This text might be better considered in the results section as it is fairly descriptive text. The discussion highlights GH-3505, GH-4859 and TZm-1353 as the best candidates for drought-prone areas, is there any further detail that can be provided about these landraces linked to the flowering / grain-filling stages stated as high risk in the introduction, to delayed silking or increased anthesis-silking interval, reduced kernel set as also given in the introduction page 2 lines 40 - 49 and the base index described in the methods. What aspects of plant physiology or morphology in reference to the published literature makes them potentially suitable? The identification of the three landraces is an interesting part of the research but would benefit greatly from some additional discussion of the characteristics that make them potentially suitable which is currently missing.
Author Response
There are no more abbreviations in the revised MS and all abbreviations have been properly defined in the main text. Please, the correct abbreviation is DSHS (combined drought and heat stress) and not DSHTS. This has been corrected in the text. The significant points regarding the genetic and phenotypic correlations has been drawn and included in the results section of the MS. The text on page 29 of the discussion line 348-359 have been included in the results section. The landrace accessions GH-3505, GH-4859 and TZm-1353 identified as outstanding under the stresses combined high grain yield potential with desirable secondary traits (reduced ASI, Tassel blast, leaf firing and increased ears per plant, and stay green characteristics as well as good Plant and ear aspects). Collectively, these features make them suitable as potential sources of alleles for improving abiotic stress tolerance. This was reflected in the high positive base index values observed for these accessions. The above have been included in the revised MS.Reviewer 5 Report
This seems to be a sound, useful contribution.
Author Response
Thank you
Round 2
Reviewer 2 Report
The authors have replied to my questions but my answer would still be the same. MS needs more improvement with more results to back their story.